# A Comparative Study of Factors Influencing Hydration Stoppage of Hardened Cement Paste

Alexander Mezhov [1,2], Daniele Kulisch [2], Antonina Goncharov [2] and Semion Zhutovsky [2,*]

1    Division of Technology of Construction Materials, Bundesanstalt für Materialforschung und -prüfung (BAM), 12205 Berlin, Germany
2    Faculty of Civil and Environmental Engineering, Technion–Israel Institute of Technology, Haifa 3200003, Israel
*    Correspondence: semionzh@technion.ac.il

**Abstract:** There is no consensus on which hydration stoppage method is optimal to preserve the microstructure and mineral composition of samples, especially considering the specific aspects of different testing methods, such as TGA, MIP, or XRD. This paper presents a quantitative comparison between the most popular hydration stoppage strategies and parameters such as the sample piece size, the soaking time in a solvent, and the type, as examined on cement paste hydrated for 7 days. It was found that the carbonation appears either for samples smaller than 2.36 mm and bigger than 4.75 mm or samples soaked in a solvent for longer than 1 h. Fast solvent replacement leads to ettringite diminution and total pore volume increase. Among others, solvent replacement with subsequent gentle heating under a vacuum was found to be the most efficient, whereas it was experimentally demonstrated that isopropyl alcohol stops hydration faster than ethanol and acetone.

**Keywords:** hydration stoppage; solvent replacement; soaking time; sample piece size

## 1. Introduction

Hydration stoppage is the common practice required to preserve the structure, mineral composition, and chemical composition of hydrated cement samples during their entire storage period [1,2]. Hydration stoppage is aimed to remove the water from a specimen. Hydrated cement paste consists of water in different states, which are classified as structural water, gel water, and free water [3]. Structural water is an integral part of hydration products. Gel water remains on the surfaces of the main hydration products (C-S-H). Free water, which is also known as capillary water, is preserved inside large pores (10 nm–100 μm) and is available for the development of hydration. Since structural and gel water are a part of the mineral composition of samples, only free water has to be removed during hydration stoppage. When choosing the method for hydration stoppage, the age of a sample has to be considered, e.g., the hydration stoppage of cementitious materials at an early age (less than 1 day) is usually more complex, since such samples are mechanically weak and consist a large amount of free water [3].

Many techniques are used for the hydration stoppage of cement paste at a later age of hydration (older than 1 day). Overall, all methods can be distinguished between a direct drying approach and a solvent exchange. During direct drying, water is evaporated or sublimated from a sample, while during solvent exchange, water in a sample is replaced by a solvent that is inert to cement and does not foster further hydration.

### 1.1. Direct Drying Method

There are three frequently used direct drying techniques: oven drying, vacuum drying, and freeze drying. During oven drying, the sample is exposed to a high temperature (60–105 °C) under atmospheric pressure [3]. However, a high temperature (105 °C) induces internal stresses, which leads to an adverse change in the microstructure and decomposition

of C-S-H and ettringite [4–8]. Additionally, an elevated temperature results in redundant carbonation [6,9], rearrangement of the pore structure [10], and formation of coarse pores (>50 nm) [11]. Moreover, there is research showing that oven drying entirely removes gel water [7]. Oven drying at 60 °C is used because it is less harmful for the mineral composition of samples than 105 °C. Yet there is evidence showing that even at 40 °C ettringite could be transformed into monosulfate [12] or decomposed [4].

Vacuum drying utilizes low pressure without elevating the temperature. It was reported that the efficiency of vacuum drying is related to the applied pressure and the duration of exposure. Placing the sample under a moderate vacuum (ca. 30 kPa) allows for the preservation of ettringite and the AFm phases and prevents carbonation. Only long treatment allows for the complete stoppage of hydration, while reducing the pressure to 200 Pa helps speed up the stoppage of hydration, but this leads to the disruption of the AFt and AFm phases [2,12].

During the freeze-drying method, the sample is immersed in liquid nitrogen, allowing for the transformation of water into ice. Later, the sample is relocated to a freeze dryer at a low temperature (−40 to −80 °C) and low pressure (0.1 to 5 Pa), resulting in the sublimation of the ice into vapor [2,3]. Nevertheless, the formation of ice leads to the appearance of cracks and damages of the microstructure [1,3,6], the failure of the C-S-H structure [13], and the decomposition of ettringite [14].

Apart from above-mentioned direct drying methods, there are a few alternatives including microwave drying, dry-ice drying (D-drying), and perchlorate drying (P-drying). Microwave drying is an accelerated oven drying technique [3]. However, these methods are considered harmful for ettringite crystals and the pore structure of samples [3,9,10,12].

It is worth noting that regardless of the type of direct drying method, in order to preserve the mineral composition and microstructure, the hydration stopping procedure is recommended to be no longer than 1 h [1,2].

### 1.2. Solvent Exchange Method

The solvent exchange method is based on the replacing of free water inside the sample with the solvent, which is inert to cement minerals. Conventionally, the procedure has two steps: immersion of the sample in a solvent and later and removal of the solvent by evaporation [2,3]. The solvent chosen for a hydration stoppage should meet the following requirements: be miscible in water; have a small molecular size to penetrate into small pores; have a low boiling point to evaporate without heating; have a low surface tension to minimize the damage to the pore structure during drying [2,11,15].

During hydration stoppage, the solvent diffuses into the structure of the sample. Therefore, the sample piece size and duration of immersion have a critical influence on the efficiency of the method.

Vast variations in the duration time of a solvent replacement can be found in the literature. The immersion periods vary from 15 min [16,17] and longer, e.g., 1 day [18], 2 days [19,20], and 4 days [21], to 21 days [11].

The time required for a sufficient solvent replacement can be calculated using the following formula: $t_E = a^2/D$, where $a$ is the characteristic dimension, and $D$ is the diffusivity [3]. The diffusivity determines the rate of penetration of a solvent into the pore structure. Hence, the diffusivity is directly related to the type of solvent and the pore structure of a sample. Based on the above equation, the sample piece size also plays a major role. It can also be concluded that the water-to-cement ratio, together with the paste volume fraction, are the parameters that significantly influence the immersion time [22]. A huge variety of piece sizes and shapes can be found in the literature [18,19,23–25]. Yet it was pointed out that the optimal size for a solvent replacement using IPA is 1 mm [26].

To facilitate an efficient solvent exchange, the solvent-to-sample ratio has to be high. Solvent-to-sample ratios from 10:1 to 100:1 are referred to in the literature. However, a lower solvent-to-sample ratio can be used if the solvent is renewed frequently [3].

The most common solvents being used for the hydration stoppage are methanol, ethanol, acetone, and isopropyl alcohol (IPA) [2]. Methanol is known to facilitate a reaction with C-S-H [6], portlandite [23,27], and cement minerals, which results in a reduction in the amount of chemically bound water [6,18,19]. Additionally, methanol destroys ettringite and the AFm phases [9,28] and causes a pore-coarsening effect that alters the microstructure of the sample [19]. Ethanol is considered less harmful for the mineral composition and microstructure of samples than methanol [2]. Yet a significant reduction in ettringite amount was observed by several researchers [2,12,28]. Application of acetone evokes the reaction with portlandite, which leads to the condensation of aldol [22] and, further, the underestimation of portlandite and overestimation of calcite [2,15]. A number of studies recommend avoiding the use of acetone as a solvent, since it alters the results of thermal analysis [29]. However, the microstructures of samples immersed in acetone were preserved better than those when using direct drying methods [1]. Most of the recent studies found that IPA was the most suitable solvent for hydration stoppage [3,4,15,23]. Even though IPA was observed to react with portlandite, however, its effect is much weaker than that of the other solvents [2,15]. The application of IPA reduces the number of coarse pores (>50 nm) and increases the number of small pores (<25 nm) in comparison to direct drying methods [11]. A solvent exchange using IPA followed by vacuum drying is considered as one of the most delicate methods of hydration stoppage [30]. Nevertheless, some studies have shown that rapid drying under a vacuum results in damage to the microstructure [15]. Instead, a second solvent exchange using a less polar solvent possessing a lower boiling point (e.g., diethyl ether) can be recommended [2,4,15,31].

An ideal solvent to be used for the hydration stoppage of cementitious materials should have a minimal chemical reaction with both cement and hydration products and have a critical point below 40 °C [16]. For instance, trifluoromethane (R23) has a low critical point (25.7 °C and 4.82 MPa) and is expected to be inert to cement [16]. Since R23 is a nonpolar liquid, it is immiscible with water, so an intermediate solvent is needed that is miscible with both water and R23, such as isopropyl alcohol [14,16,17].

As it is demonstrated in the introduction, there many studies aimed to find a method for stopping hydration, which is less harmful for the samples and, at the same time, is predictable and reliable. There are many aspects that have a critical impact on the stoppage of hydration, i.e., the sample piece size, the type of solvent, and the application of additional treatment. Yet there is an important aspect, which is being overlooked, i.e., the combination of all the aforementioned factors and their influence on the stopping of hydration. Thus, the current paper provides a systematic study of the most critical factors influencing and their mutual effect on the stopping of the hydration of a hardened cement paste.

There is no consensus among the scientific community regarding the effect of the principal aspects on hydration stoppage, and many papers contradict each other. The current research aims to elucidate the influence of the sample piece size, the duration of the solvent replacement, the solvent type, and additional treatments, such as the vacuum drying and freeze drying of the mineral composition and the microstructure of the hardened cement paste.

## 2. Materials and Methods

### 2.1. Materials

Standard cement of CEM I 52.5N type supplied by Nesher Israel Cement Enterprises was used for this study. Its chemical and mineral compositions are presented in Tables 1 and 2, respectively. Loss on ignition at 950 °C of this cement was 2.58%, and specific surface area was 4250 cm$^2$/g. Solvents for hydration stoppage were chemically pure isopropyl alcohol, acetone, and ethyl alcohol. Particle size distribution and heat flow evolution of studied cement is given in Figure 1.

**Table 1.** Chemical composition of cement (wt.-%), obtained by ICP analysis.

| CaO | SiO$_2$ | Al$_2$O$_3$ | Fe$_2$O$_3$ | MgO | TiO$_2$ | K$_2$O | Na$_2$O | P$_2$O$_5$ | Mn$_2$O$_3$ | SO$_3$ | IR | FL | LOI |
|---|---|---|---|---|---|---|---|---|---|---|---|---|---|
| 62.16 | 19.02 | 5.42 | 3.82 | 1.31 | 0.53 | 0.37 | 0.22 | 0.40 | 0.05 | 2.48 | 0.76 | 2.80 | 2.93 |

**Table 2.** Structures used for Rietveld refinement and mineralogical composition of cement according to the International Centre for Diffraction Data (ICDD) and Inorganic Crystal Structure Database (ICSD).

| Phase | Formula | Database | Code | wt.% Dry Cement |
|---|---|---|---|---|
| Alite | C$_3$S | ICDD | 04-011-1393 | 62.00 |
| Belite | C$_2$S | ICDD | 04-007-9746 | 14.16 |
| Aluminate cubic | C$_3$A | ICSD | 98-000-1841 | 5.86 |
| Ferrite | C$_4$AF | ICDD | 04-007-5261 | 12.14 |
| Calcite | C$\bar{C}$ | ICSD | 98-019-1857 | 1.96 |
| Bassanite | C$\bar{S}$H$_{0.5}$ | ICSD | 98-038-0286 | 2.80 |
| Portlandite | CH | ICDD | 04-006-9147 | 1.65 |
| Arcanite | K$\bar{S}$ | ICDD | 00-024-0703 | 1.23 |
| Aphthitalite | K$_3$Na$\bar{S}_4$ | ICDD | 00-020-0928 | 0.56 |
| Ettringite | C$_6$A$\bar{S}_3$H$_{32}$ | ICDD | 01-073-6239 | – |
| Monosulfate | C$_4$A$\bar{S}$H$_{12}$ | ICDD | 04-013-3303 | – |
| Hydrogarnet | C$_3$AH$_6$ | ICDD | 04-017-1504 | – |
| Hemicarbonate | C$_4$A$\bar{C}_{0.5}$H$_{12}$ | ICSD | 98-026-3124 | – |

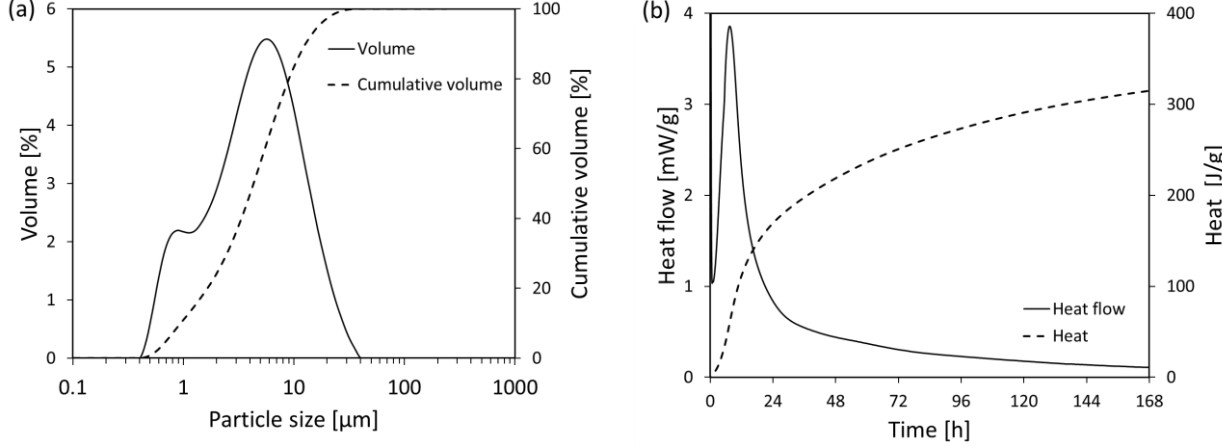

**Figure 1.** (**a**) Particle size distribution and (**b**) heat flow evolution of studied cement.

*2.2. Preparation of Samples*

Cement was mixed with water in a pan mixer for 5 min and then cast in 25 mm cube molds. The water-to-cement ratio was 0.40. After 24 h of hydration in sealed molds, samples were demolded and immersed in water for 6 days. The hydration was stopped at 7 days by the different methods. Then, samples were crushed by hand using pestle and mortar and sieved. Particles that passed a sieve of 4.75 mm and were retained on a sieve of 2.36 mm were selected for the analyses. Detailed descriptions of sample preparation, depending on the studied parameter, are given below.

- Duration of solvent replacement

Samples were immersed in isopropyl alcohol (IPA) for $^1/_4$, $^1/_2$, 1, and 24 h. Immediately after removing the samples from the solvent, they were placed in a vacuum oven at 40 °C for 3 h. The ratio of solvent to the sample mass was set at 20 for all the procedures.

- Solvent type

Samples were placed in three plastic vials and immersed in one of the solvents: isopropyl alcohol (IPA), acetone (Ac), or ethyl alcohol (Eth) for $^1/_2$ h divided into 2 stages.

In the first stage, the sample was immersed in solvent for 1/4 h. Then, the solvent was replaced by a second batch of the same solvent, for another 1/4 h. Immediately after removing the samples from the solvent, they were placed in a vacuum oven at 40 °C for 3 h.

- Sample piece size

Samples were crushed by hand using pestle and mortar and sieved. Samples with the following sizes were selected for the analyses: (1) uncrushed sample 25 × 25 × 25 mm; (2) particles passed 9 mm sieve and retained on a 4.75 mm sieve; (3) particles passed 4.75 mm sieve and retained on a 2.36 mm sieve; (4) particles passed 2.36 mm sieve and retained on a 1.2 mm sieve. After crushing, samples were immersed with isopropyl alcohol (IPA) for 1/2 hour (solvent was replaced after 1/4 hour). Immediately after removing the samples from the solvent, they were placed in a vacuum oven at 40 °C for 3 h.

- Drying and freezing

Samples were treated by three different methods. The first part was placed directly in a vacuum chamber for 3 h (V) at 21 °C, the second part was placed in a vacuum chamber for 3 h (V + O) at 40 °C, and the third part was immersed in liquid nitrogen for 5 min and then moved in a vacuum chamber for 3 h (F).

An example of coding of different samples is given in Figure 2 detailed description of the samples is presented in Table 3.

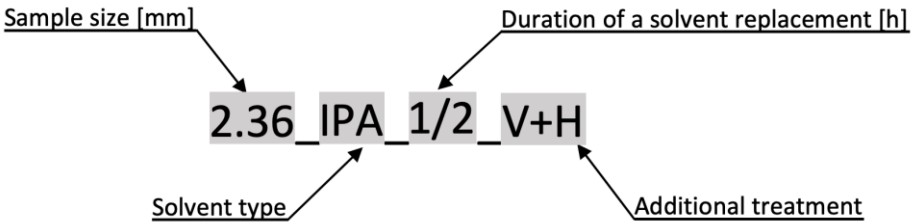

**Figure 2.** Code for samples applied in the study.

**Table 3.** Detailed description of the studied samples.

| Sample Name | Sample Piece Size (mm) | Solvent Type | Duration of a Solvent Replacement (h) | Additional Treatment |
|---|---|---|---|---|
| Influence of the sample piece size | | | | |
| 1.20_IPA_1/2_V+H | 1.2 | IPA | 1/2 | vacuum for 3 h at 40 °C |
| 4.75_IPA_1/2_V+H | 4.75 | IPA | 1/2 | vacuum for 3 h at 40 °C |
| 25.0_IPA_1/2_V+H | 25 | IPA | 1/2 | vacuum for 3 h at 40 °C |
| Influence of the duration of a solvent replacement | | | | |
| 2.36_IPA_1/4_V+H | 2.36 | IPA | 1/4 | vacuum for 3 h at 40 °C |
| 2.36_IPA_1_V+H | 2.36 | IPA | 1 | vacuum for 3 h at 40 °C |
| 2.36_IPA_24_V+H | 2.36 | IPA | 24 | vacuum for 3 h at 40 °C |
| Influence of the solvent type | | | | |
| 2.36_IPA_1/2_V+H * | 2.36 | IPA | 1/2 | vacuum for 3 h at 40 °C |
| 2.36_Ac_1/2_V+H | 2.36 | acetone | 1/2 | vacuum for 3 h at 40 °C |
| 2.36_Eth_1/2_V+H | 2.36 | ethanol | 1/2 | vacuum for 3 h at 40 °C |
| Influence of drying and freezing | | | | |
| 2.36_V | 2.36 | – | – | vacuum for 3 h at 21 °C |
| 2.36_V+H | 2.36 | – | – | vacuum for 3 h at 40 °C |
| 2.36_F+V | 2.36 | – | – | freezing with liquid nitrogen for 5 min and vacuum for 3 h at 21 °C |

* The sample is defined as a reference for comparison with other treatments and appears in red color in the following figures.

### 2.3. Experimental Methods

### 2.3.1. Particles Size Distribution

The analysis of particle size distribution of cement was performed by means of a Malvern Mastersizer 3000 laser diffractometer. The measurement was performed in liquid media of isopropyl alcohol (IPA). The frequency and cumulative particle size distribution of cement are given in Figure 1.

### 2.3.2. Thermo-Gravimetric Analysis

Thermo-gravimetric analysis (TGA) was performed using a Netzsch STA 449 F5 Jupiter instrument at a heating rate of 20 °C/min in alumina crucibles. Samples were subjected to heating from 40 °C to 1000 °C under nitrogen atmosphere purged at the rate of 50 mL/min.

The samples for TGA were manually ground, after hydration stoppage, using mortar and pestle. The samples were completely ground into powder to pass the 45-micron sieve. The powder sample mass was around 50 mg for all TGA tests.

Since all the hydration stoppage methods used in this research are intended to remove only free capillary water, ideally the bound water lost in the process of the TGA test includes both gel and chemically bound water. The ultimate chemically bound water in cement-hydration products can be expressed based on its mineral composition [32]:

$$W_{ch} = a_w(C_3S) + b_w(C_2S) + c_w(C_3A) + d_w(C_4AF) \tag{1}$$

where $W_{ch}$ is chemically bound water. In the current work, we use the following coefficients, $a_w = 0.229$, $b_w = 0.199$, $c_w = 0.522$, and $d_w = 0.109$, for chemically bound water. Coefficients of this equation vary slightly from publication to publication [32–36]. Since the cementitious gel is formed only by alite and belite, the amount of gel water can be calculated based on the stoichiometry of their hydration reactions as follows:

$$C_3S + 5.3 \cdot H \rightarrow C_{1.7}SH_4 + 1.3 \cdot CH \tag{2}$$

| Molar mass, (g/mole) | 228.33 | 95.48 | 227.4 | 96.32 | |
|---|---|---|---|---|---|
| $W_{ch}$, (g/mole of $C_3S$) | | | 28.83 | + 23.42 | $\approx 52.25$ |
| $W_{gel}$, (g/mole of $C_3S$) | | | 43.24 | | |
| $W_{ch}$, (g/g of $C_3S$) | | | 0.126 | + 0.103 | $\approx 0.229$ |
| $W_{gel}$, (g/g of $C_3S$) | | | 0.189 | | |

$$C_2S + 4.3 \cdot H \rightarrow C_{1.7}SH_4 + 0.3 \cdot CH \tag{3}$$

| Molar mass, (g/mole) | 172.25 | 77.47 | 227.49 | 22.23 | |
|---|---|---|---|---|---|
| $W_{ch}$, (g/mole of $C_2S$) | | | 28.83 | + 5.40 | $\approx 34.23$ |
| $W_{gel}$, (g/mole of $C_2S$) | | | 43.24 | | |
| $W_{ch}$, (g/g of $C_2S$) | | | 0.1673 | + 0.0314 | $\approx 0.199$ |
| $W_{gel}$, (g/g of $C_2S$) | | | 0.251 | | |

In the above calculations, the stoichiometric formula of cementitious gel in Equations (2) and (3) contains 4 moles of water per mole of cementitious gel, which includes approximately 1.6 moles of chemically bound water and 2.4 moles of gel water. Of course, the water in portlandite also has to be included in the amount of chemically bound water. Accordingly, the gel water can be calculated using the following formula:

$$W_g = a_g(C_3S) + b_g(C_2S) \tag{4}$$

where $W_g$ is total gel water, and $a_g = 0.189$ and $b_b = 0.251$ are coefficients of gel water for alite and belite, respectively. Accordingly, considering the mineralogical composition of cement given in Table 2, the chemically bound water ($W_{ch}$) is 21.4%, and gel water ($W_g$) is 15.3%, by initial weight of anhydrous cement. This gives us total bound water ($W_{ch+g}$) as 36.7% by weight of cement. Besides the content of bound water, the most important

parameter of thermogravimetric analysis is the temperature range for its quantification. One of the widely accepted approaches in the past was measuring mass loss between 105 and 1000 °C [37]. However, this was mainly applicable to oven-dried samples. As shown previously, drying in an oven at 105 °C removes some of the chemically bound and gel water, reducing the measured bound water value. Since the hydration-stopping procedures described above, which involved drying at 40 °C, are supposed to remove all the free (capillary) water from the sample, it makes sense to measure the mass loss for bound water starting from 40 °C. However, there is no consensus about the final temperature. Some researchers suggest the final temperature for bound water by mass loss step is between 500–550 °C. This excludes the $CO_2$ decomposition step that takes place approximately between 600 and 800 °C. However, one can argue that the source of $CO_2$ is calcite, which is a product of the carbonation of portlandite. Portlandite, in turn, is a product of cement hydration. So it should be taken into account by converting calcite to portlandite or, since the stoichiometric ratio of carbon dioxide and water to lime in both calcite and portlandite is 1:1, mass loss step of calcite can be converted to mass loss of chemically bound water by the molar ratio of water to carbon dioxide. Of cause, the initial content of calcite in cement should be considered in this procedure.

Portlandite content ($m_{Ca(OH)_2}$) was determined as described by Taylor [34] based on the weight loss as "Marsh" step (WLMS), which is also referred to as a tangential method [38], with the start of the tangent slope at 405 °C and the end at 590 °C ($WLMS_{Ca(OH)_2}$):

$$m_{Ca(OH)_2} = \frac{WLMS^p_{Ca(OH)_2} - WLMS^c_{Ca(OH)_2}}{m^c_i \, (1 + w/c)} \cdot \frac{M_{Ca(OH)_2}}{M_{H_2O}} \tag{5}$$

where $WLMS^p_{Ca(OH)_2}$ and $WLMS^c_{Ca(OH)_2}$ are WLMS of hydrated cement paste and cement, respectively; $M_{Ca(OH)_2}$ and $M_{H_2O}$ are molar masses of portlandite and water that are equal to 74 and 18 g/mol, respectively; and $m^c_i$ is the initial mass fraction of unhydrated cement in the paste, which is determined as follows:

$$m^c_i = \frac{1 - WL^p_{40-950°C}}{1 - WL^c_{40-950°C}} \tag{6}$$

Here, $WL^p_{40-950°C}$ and $WL^c_{40-950°C}$ are weight loss between 40 and 950 °C of hydrated cement paste and unhydrated cement, respectively. Similarly, calcite content was determined based on the weight loss as "Marsh" step between 605 and 780 °C of hydrated cement paste ($WLMS^p_{CaCO_3}$) and cement ($WLMS^c_{CaCO_3}$) as follows:

$$m_{CaCO_3} = \frac{WLMS^p_{CaCO_3} - WLMS^c_{CaCO_3}}{m^c_i \, (1 + w/c)} \cdot \frac{M_{CaCO_3}}{M_{CO_2}} \tag{7}$$

where $M_{CaCO_3}$ and $M_{CO_2}$ are molar masses of calcite and carbon dioxide equal to 100 and 44 g/mol, respectively. Note that the content of hydration products is usually expressed relative to the initial mass of cement paste, which is calculated as the initial mass of cement and added mixing water in accordance with the w/c ratio.

Accordingly, in this work, we calculated the degree of hydration ($\alpha$) based on bound water that was determined based on loss of ignition at two different temperatures of 550 and 950 °C as follows:

$$\alpha^{550°C} = \frac{WL^p_{40-550°C} - WL^c_{40-550°C} + \left(WLMS^p_{CaCO_3} - WLMS^c_{CaCO_3}\right)\left(\frac{M_{H_2O}}{M_{CO_2}}\right)}{m^c_i \, W_{ch+g}} \tag{8}$$

$$\alpha^{950°C} = \frac{WL^p_{40-950°C} - WL^c_{40-950°C} + \left(WLMS^p_{CaCO_3} - WLMS^c_{CaCO_3}\right)\left(\frac{M_{H_2O}}{M_{CO_2}} - 1\right)}{m^c_i \, W_{ch+g}} \tag{9}$$

While the former mass loss includes only bound water, in the latter, the calcite step is also included. For this reason, the member of the equation that is responsible for the conversion of carbon dioxide mass loss of calcite to water mass loss of portlandite contains an additional "−1" member. Note that for calculating the degree of hydration properly, the bound water has to be expressed relative to the initial mass of cement, which is assessed as ignited mass corrected to account for the loss on ignition of cement. $W_{ch+g}$ in Equations (8) and (9) is the total bound water that was previously calculated as 36.7% by weight of cement.

The content of cementitious gel in hydrated cement paste, which otherwise is called calcium silicate hydrate (C-S-H), can be evaluated based on bound water. Assuming that by excluding from the total bound water the water chemically bound to crystalline phases, we obtain water bound to C-S-H only:

$$W_{csh} = a_{csh}(\mathrm{C_3S}) + b_{csh}(\mathrm{C_2S}) \tag{10}$$

where the coefficients $a_{csh}$ and $b_{csh}$ can be obtained from Equations (2) and (3) using both chemically bound water and gel water related to C-S-H only: $a_{csh} = 0.189 + 0.126 = 0.315$ and $b_{csh} = 0.251 + 0.167 = 0.418$, which consider contents of alite and belite results in $W_{csh} = 25.5\%$. Accordingly, C-S-H content can be calculated as follows:

$$
m_{CSH}^{550°C} = \left( WL_{40-550°C}^{p} - WL_{40-550°C}^{c} - \left( WLMS_{AFt}^{p} - WLMS_{AFt}^{c} \right) - \\ - \left( WLMS_{Ca(OH)_2}^{p} - WLMS_{Ca(OH)_2}^{c} \right) \right) / \left( m_i^c \left( 1 + w/c \right) W_{csh} \right) \tag{11}
$$

$$
m_{CSH}^{950°C} = \left( WL_{40-950°C}^{p} - WL_{40-950°C}^{c} - \left( WLMS_{AFt}^{p} - WLMS_{AFt}^{c} \right) - \\ - \left( WLMS_{CaCO_3}^{p} - WLMS_{CaCO_3}^{c} \right) \right) / \left( m_i^c \left( 1 + w/c \right) W_{csh} \right) \tag{12}
$$

where $WLMS_{AFt}^{p}$ and $WLMS_{AFt}^{c}$ are the weight loss as "Marsh" step due to AFt of hydrated cement paste and unhydrated cement, respectively, with the start of the tangent slope at 40 °C and the end at 300 °C. Equations (11) and (12) give a rough estimation of C-S-H, assuming that if all weight losses due to crystalline phases are excluded, the only weight loss left is due to loss of water bound to the amorphous C-S-H phase. The crystalline phases peaks in DTG are related to AFt, portlandite, and calcite. To assess this assumption, C-S-H content by TG will be compared with the results obtained by XRD analysis. Note that the C-S-H content is expressed relative to the initial mass of cement paste.

### 2.3.3. X-ray Diffraction

The samples for X-ray diffraction were manually ground, after hydration stoppage, using mortar and pestle. The samples were completely ground into powder to pass the 45-micron sieve, and the mean particle size is 20 μm. Based on the mean particle size, the particle statistic error, according to [2], is 8%. The powder sample was backloaded into a round sample holder with an open scanning area of 27 mm diameter.

The X-ray diffraction analysis was performed using a Malvern PANAlytical EMPYREAN X-ray diffractometer with the following configuration: an X-ray source was CuKα1,2 (λ = 1.5408 Å), with an X-ray generator operated at a voltage of 45 kV and a current of 40 mA; goniometer radius was 240 mm; the incident beam optics included 10 mm mask, 0.04 rad Soller slit along with 1/4 ° divergence, and 1° anti-scatter fixed slits; the diffracted beam optics consisted of 8 mm anti-scatter fixed slit and 0.04 rad Soller slit. The spinning sample stage was used with 1 revolution per second during the scanning process. The detector was a PIXcel 3D detector used in 1D continuous scan mode. The scan was performed using Bragg–Brentano geometry, between 10 and 70°2θ for unhydrated cement and between 5 and 70°2θ for hydrated cement paste. In addition, the time step, step size, and total time were different between cement and cement paste. For cement paste, a timestep of 80.32 s along with a goniometer step size of 0.013°2θ were used, resulting

in a total measurement time of 25.22 min. For cement paste, one program for all samples was applied because the same intensity and range of scanning angles were used. The quantitative analysis was performed by means of Rietveld refinement using HighScore Plus software. The average weighted R profile ($R_{wp}$) was 3.8992, with a standard deviation of 0.2805, and the average goodness of fit was 1.4373, with a standard deviation of 0.1110.

The description of phases used for Rietveld refinement, their database source, and reference code are given in Table 2. The content of clinker minerals was normalized by the weight of cement, which was evaluated based on ignited mass, as described in the Section 2.3.2. The degree of hydration was calculated based on the residual content of cement minerals as follows:

$$\alpha^{XRD} = 1 - \frac{m_{C_3S} + m_{C_2S} + m_{C_3A} + m_{C_4AF}}{m_i^c \left( m_{C_3S}^c + m_{C_3S}^c + m_{C_3S}^c + m_{C_3S}^c \right)} \quad (13)$$

where $m_i^c$ is the initial mass fraction of unhydrated cement in the paste calculated according to Equation (6), $m_{C_3S}, m_{C_2S}, m_{C_3A}$, and $m_{C_4AF}$ are mass fractions on four main cement minerals in cement paste as determined by XRD including amorphous phase, and $m_{C_3S}^c, m_{C_3S}^c, m_{C_3S}^c$, and $m_{C_3S}^c$ are mass fractions on four main cement minerals in unhydrated cement (% wt.), which are given in Table 2 (this gives 94.16% wt. in total for cement used in the current research).

The content of hydration phases was normalized by the weight of the initial cement paste, as described in the Section 2.3.2. The analysis of the amorphous phase was performed using the external standard method. Fully crystalline alumina ($\alpha$-$Al_2O_3$) was used as a standard. The C-S-H content was calculated as the content of the amorphous phase normalized to the weight of the initial cement paste as follows:

$$m_{CSH}^{XRD} = \frac{m_{amorphous}}{m_i^c \left(1 + w/c\right)} \quad (14)$$

where $m_{amorphous}$ is the content of the amorphous phase (% wt.), as determined by XRD using an external standard.

### 2.3.4. Isothermal Calorimetry

Calorimetric measurements were performed using the TAM Air Isothermal Calorimeter (TA instruments). The measurement precision of the heat flow was $\pm 4$ µW at a 95% confidence interval. Eight independent samples consisting of cement (3.6 g) and deionized water (1.44 g) were prepared and mixed by vibration at 60 rpm for 20 s. The samples were then introduced into the calorimeter, which measured the heat flow for 7 days, providing 6 measurements per minute. The average heat flow and total heat of hydration are given in Figure 1. All the tests were conducted at 21 °C.

Total heat was used to calculate the degree of hydration for the comparison with the hydration stoppage methods. The first 30 min of the measurement were disregarded because of thermal equilibration. Accordingly, the average total heat at 7 days (168 h) was $H_{7d} = 314$ J/g. The degree of hydration was calculated based on the ratio of the measured heat and ultimate heat of hydration ($H_t$) [34]:

$$H_t = a_h(C_3S) + b_h(C_2S) + c_h(C_3A) + d_h(C_4AF) \quad (15)$$

where $H_t$ is cumulative heat (J/g). Using mass fractions of the clinker minerals from Table 2 and the following coefficients, $a_h = 517$, $b_h = 262$, $c_h = 1672$, and $d_h = 418$ J/g [34], the ultimate heat of hydration of 506.4 J/g was obtained.

After 7 days, 6 samples were opened, and a solvent was added. The solvents used in this test included isopropyl alcohol, ethanol, and acetone with two duplicate samples for each. The samples were sealed and inserted back for 7 days. For reference, a pair of samples were left to continue the hydration inside the calorimeter.

2.3.5. Mercury Intrusion Porosimetry (MIP)

Cumulative pore volume and pore size distribution of hardened pastes were determined using a Quantachrome PoreMaster-60 apparatus. Mercury surface tension of 480.00 erg/cm$^2$ and contact angle of 140° were used in calculations. Two types of pressure were used: low (from 1 to 24 psi) for the determination of coarse pores and high (from 14 to 60,000 psi) for the determination of fine pores. The tests were performed on the granular samples that were prepared as described above. MIP dilatometer was filled with a sample to achieve a high number of pieces of the sample, which increases its surface-to-volume ratio and helps to reduce boundary effects.

## 3. Results and Discussion

### 3.1. Thermal Analysis

The results of the thermal analysis are presented in Figure 3a–d. It can be seen in Figure 3a that the piece size of the sample during hydration stopping has a notable influence on the amount of the AFt phase, as determined by thermal analysis. The larger the piece size is, the higher the effective content of the AFt and C-S-H phases is. However, the reference (piece size 2.36 mm) demonstrated the lowest value. Calcite content (Cc) shows the same trend. Apparently, the increased piece size of the sample promotes the excess carbonation of the sample. Yet the amount of the portlandite phase is similar for all specimens. Interestingly, the specimens with the largest piece size (25.0_IPA_1/2_V+H) and the smallest piece size (1.2_IPA_1/2_V+H) exhibited a notable increase in calcite content. In both cases, the presence of a higher calcite content indicates carbonation.

Evidently, soaking time also affected AFt content (see Figure 3b). Up to 30 min, the influence is insignificant, whereas longer soaking results in the augmentation of the AFt peak. This is owing to the known ability of ettringite to strongly absorb different solvents [39]. Consequently, this prompts an apparent increase in the ettringite DTG peak in the sample immersed in ethanol, as shown in Figure 3c. The calcite content is not related to the duration of the solvent replacement nor the content of the portlandite.

The influence of the solvent type on the thermal analysis is presented in Figure 3c. As shown previously [3,39], an increase in the AFt peak is related to the adsorption of ethanol by ettringite, while acetone and IPA show less influence. Apart from the notable difference in the AFt phase, the other peaks on the DTG diagram are very similar.

Applying various methods of hydration stoppage by direct drying exerts a prominent change in the TG analysis (Figure 3d). It is worth noting the essential distinction between two specimens: 2.36_V and 2.36_V+H. In fact, the only difference between them is the temperature of treatment under vacuum (see Table 3). This is 21 and 40 °C, respectively. However, the C-S-H peak at the DTG diagram is significantly reduced when the treatment temperature is 40 °C, while the CH and Cc are barely different. The application of a solvent in a combination of a vacuum and heat slightly reduces the C-S-H peak. Unless the sample is directly exposed to a vacuum, the sample subjected to freezing coupled with a vacuum shifts the AFt peak to a lower temperature and additionally reduces the amount of calcite that is formed.

Overall, we can summarize the thermal analysis results by stating that the method chosen for the hydration stoppage mostly influences the stability of the AFt phases and carbonation.

### 3.2. Mercury Intrusion Porosimetry

In Figure 4a–d, the influence of the studied factors on the pore structure of the samples is shown. The total pore volume, critical pore radius, and threshold pore radius can be identified in these figures. Considering the effect of piece size, the total pore volume and pore entry radius are almost identical for all samples, whereas the sample with the biggest piece size exhibited a much lower total pore volume and a bigger pore entry radius (Figure 4a), which are related to the insufficient removal of water and collapse of pore structure during the drying stage.

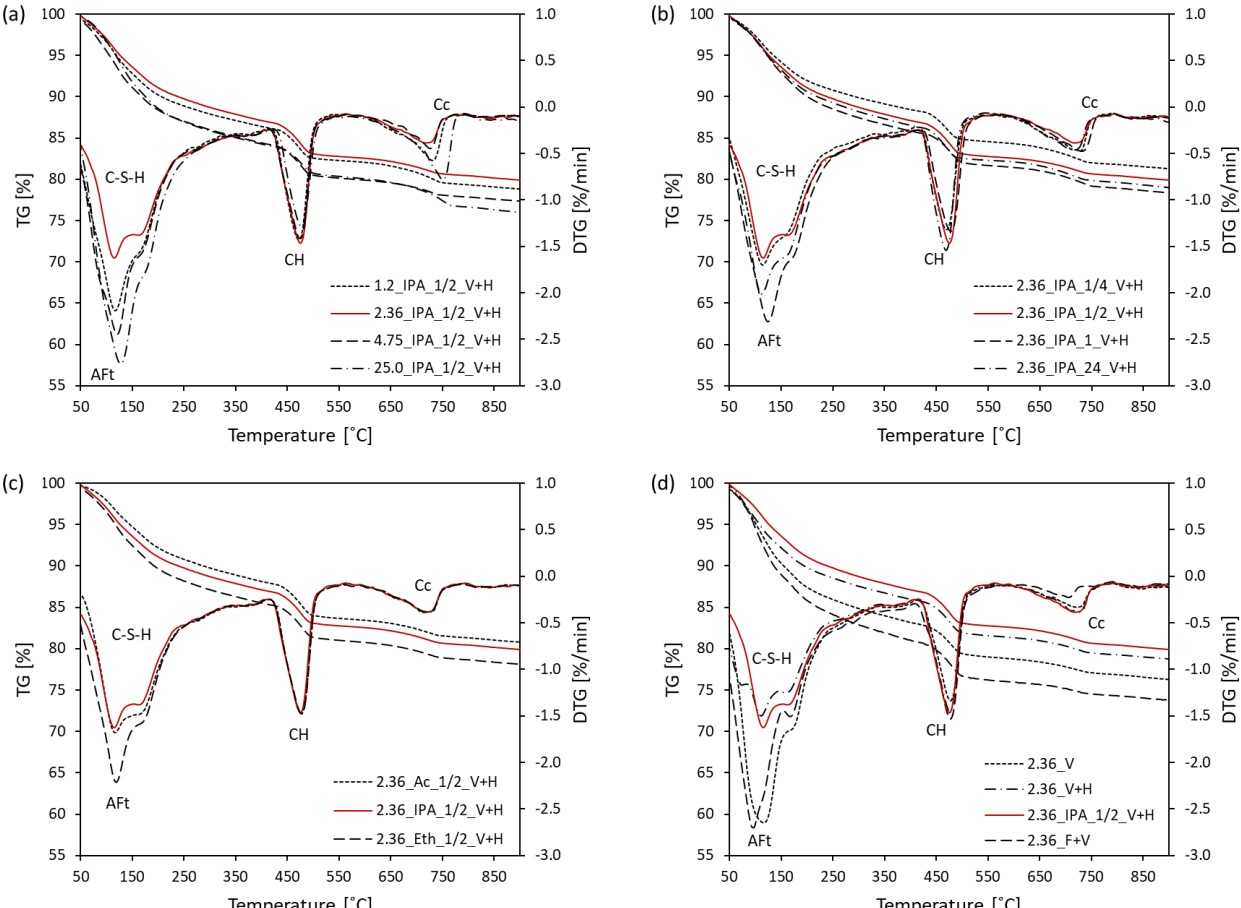

**Figure 3.** Thermogravimetric (TG) and the first derivative of temperature for thermogravimetry (DTG) curves of cement paste after 7 days of hydration showing the influence of the studied factors: (**a**) sample piece size; (**b**) duration of a solvent replacement; (**c**) solvent type; (**d**) method of hydration stoppage.

Soaking time had no significant impact on the total pore volume (Figure 4b), with the exception being the sample immersed in IPA for the shortest time, which was also a result of incomplete water removal by the solvent. The pore entry radius is the smallest for the sample subjected to the solvent for 1/2 h, while the biggest value is demonstrated by the sample immersed for 1/4 h.

The effect of the solvent type is shown in Figure 4c. According to these results, IPA can be considered as the most suitable solvent, though ethanol showed very close results. For both total pore volume and the pore entry radius, IPA demonstrated the lowest values, i.e., the coarsening effect on the pore structure of the sample was minimized.

The comparison solvent replacement with direct drying methods is shown in Figure 4d. Samples that were directly placed under vacuum exhibited a coarsening of the pore structure, with a slight increase in total pore volume and a considerable increase in pore entry radius, in comparison to the solvent replacement. The MIP of the samples subjected to a vacuum with additional heating as well as those exposed to freezing followed by vacuum drying provided evidence for the collapse of pore structure. The solvent replacement followed by vacuum drying at 40 °C (2.36_IPA_1/2_V+H) revealed the best preservation of the pore structure.

### 3.3. X-ray Diffraction

The influence of the sample piece size on the mineral phase composition is presented in Figure 5. The content of alite and ferrite remained constant regardless of the piece size, while the belite content is notably lower in the 25 mm sample. The C-S-H content was

similar for the samples with a piece size of 1.2 and 2.36 mm, while samples with a bigger piece size showed a higher amount of C-S-H. The DoH and the contents of ettringite and hydrogarnet gradually increased with the piece size of the sample. However, the content of ettringite in the piece size of 25 mm was notably lower than that of smaller sizes. The amount of portlandite (CH) increased with sample piece size. The content of the AFm phase was similar for piece sizes of 1.2 and 2.36 mm, while the bigger samples showed a reduced content of AFm.

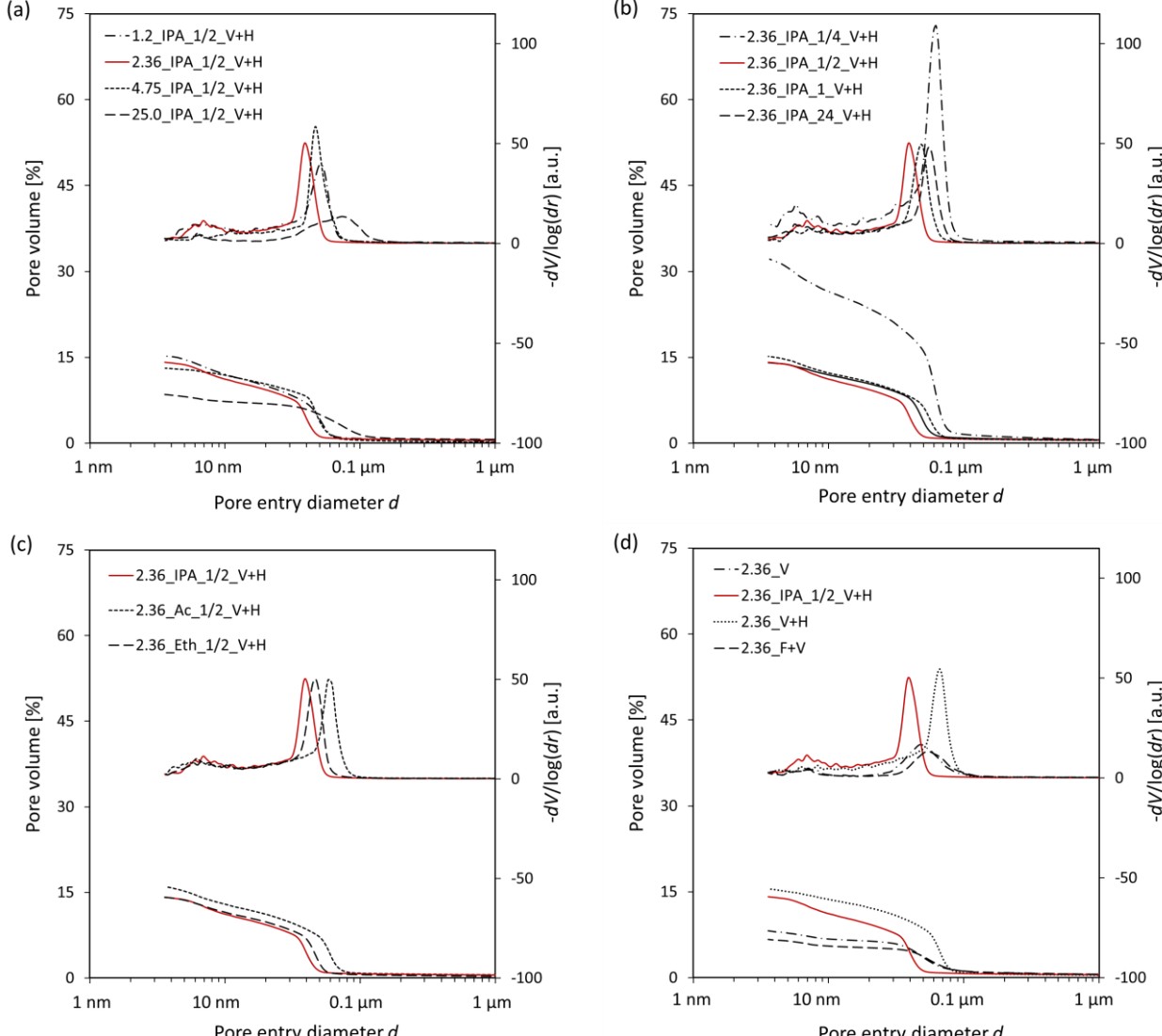

**Figure 4.** Mercury intrusion porosimetry (MIP) curves of cement paste after 7 days of hydration, showing the influence of the studied factors on the pore structure. (**a**) Different sample piece size; (**b**) different duration of a solvent replacement; (**c**) different solvent type; (**d**) different method of hydration stoppage.

DoH was slightly higher for a piece size of 4.75 mm and significantly higher for a piece size of 25 mm. It should be noted that the effect of sample piece size on residual cement minerals, portlandite, and C-S-H formation was consistent with the DoH. Thus, it can be concluded that the piece size affected the hydration stoppage, starting from 4.75 mm. A higher carbonation level was also observed in the large piece size of 25 mm. These results are consistent with the thermal analysis (see Figure 3a). The ettringite content observed using XRD was similar for the two samples with a small piece size. However, a sample

with a bigger piece size showed an increase in ettringite contents, while the biggest sample exhibited a reduced amount of ettringite.

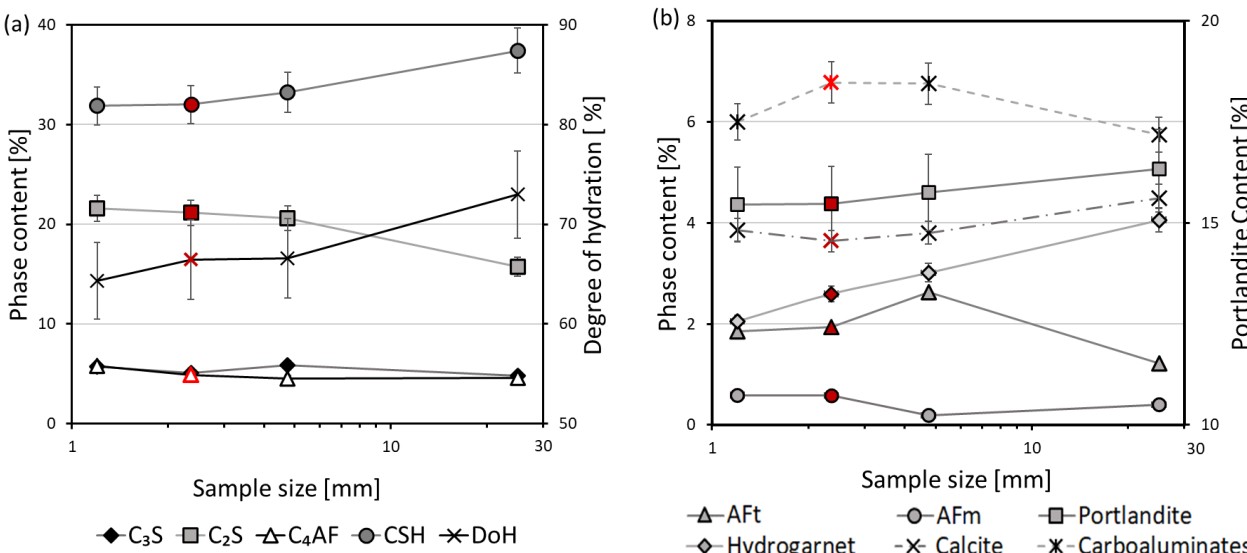

**Figure 5.** Influence of the sample piece size on: (**a**) content of cement minerals, C-S-H, and degree of hydration (calculated by TG); (**b**) content of products of hydration. The reference points for 2.36_IPA_1/2_V+H are marked in red.

The influence of the duration of solvent replacement on the hydration stoppage is shown in Figure 6. It can be seen that the duration of soaking in the solvent did not significantly affect the content of the cement minerals, i.e., alite, ferrite, and belite. The immersion duration also did not significantly affect the DoH or the C-S-H content, as determined by XRD. The most significant effect of the duration of the solvent replacement procedure is on carbonation. As can be seen in Figure 6b, the content of calcite considerably increased with the increase in soaking time, reaching 24 h with almost twice the content of calcite as at a soaking time of 15 min. The significant carbonation at the immersion time of 24 h also caused a significant reduction in portlandite content, which reacted with carbon dioxide turning into calcite.

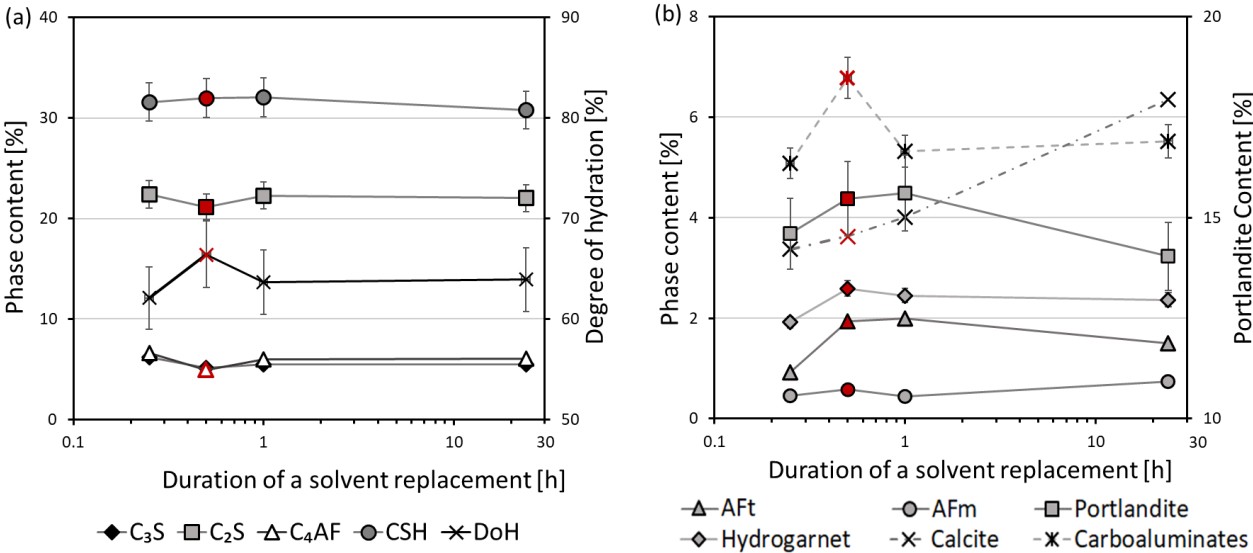

**Figure 6.** Influence of the duration of the solvent replacement on (**a**) content of cement minerals, C-S-H, and degree of hydration (calculated by TG); (**b**) content of products of hydration. The reference 2.36_IPA_1/2_V+H is marked in red.

It can be noted that the content of the aluminate phases was lower for the shortest soaking time (15 min). This indicates that a very short solvent replacement is not able to preserve the aluminate hydration phases well for further analysis, and an immersion of at least 30 min in the solvent is required for this purpose. This is consistent with the MIP results, which indicate that a short soaking time did not remove all the water, and further vacuum drying results in damage to the microstructure of the cement paste (Figure 4b).

Evidently, the XRD results reveal the carbonation of the samples after a prolonged soaking time, which is consistent with the literature data [3]. However, the XRD data did not correlate well with the thermal analysis results (see Figure 3b), which in turn showed the lower difference in portlandite content and a minor effect on carbonation, while demonstrating a considerably higher AFt content for the two highest soaking-duration times. This effect may be also explained by the adsorption of the solvent into the AFt phases and probably also into the C-S-H microstructure. Therefore, the optimum duration of a solvent replacement might be considered as a period of time at least 30 min and no more than 60 min.

In Figure 7, the effect of the solvent type on the hydration stoppage is illustrated. It can be seen that the utilization of different solvent types for the stopping of hydration had no prominent influence on the content of anhydrous clinker phases, i.e., $C_3S$, $C_2S$, and $C_4AF$. In the same fashion, the amount of C-S-H formed during hydration remained stable irrespective of the solvent type used. The DoH was slightly higher when IPA as used as a solvent. The notable influence of ethanol on the portlandite formation can be observed in Figure 7b. When IPA was utilized, the observed content of ettringite, hydrogarnet, and carboaluminates was slightly higher than in the case of a solvent exchange with acetone and ethanol.

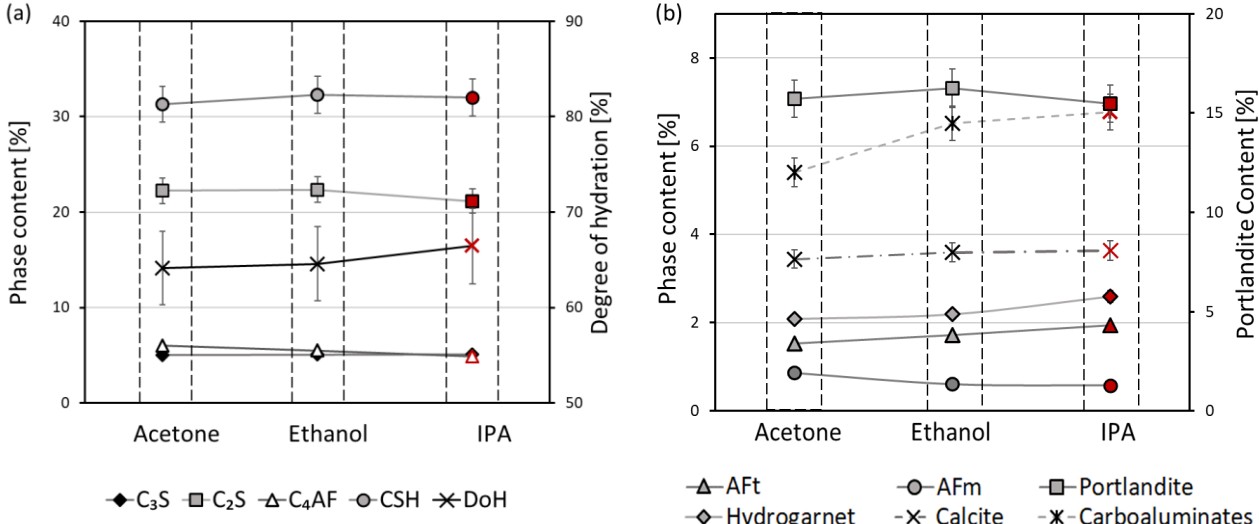

**Figure 7.** Influence of the solvent type used for the hydration stoppage on (**a**) content of cement minerals, C-S-H, and degree of hydration (calculated by TG); (**b**) content of products of hydration. The reference 2.36_IPA_1/2_V+H is marked in red.

Thermal analysis revealed the only difference between the solvents in the AFt peak in the DTG curves, while the other parts of the DTG curveswere completely identical (see Figure 3c). Interestingly, in the thermal analysis, the trend in AFt content was quite opposite to the trend in the XRD results. This indicates considerable solvent adsorption by the AFt phases in the case of ethanol and a slight solvent adsorption in the case of acetone. Both the XRD and thermal analysis showed no significant effect on carbonation.

Among the other factors investigated in this study, the most noticeable effect on the mineral phase composition was exerted by the method of stopping hydration (see Figure 8). Taking a solvent exchange with isopropyl alcohol (2.36_IPA_1/2_V+H marked by red color in Figure 8) as a reference procedure for stopping hydration, the most significant difference

was observed for freezing in liquid nitrogen. The least significant difference was observed in the case of omitting a solvent exchange and using only a vacuum oven for removing the free water from the hydrating cement paste. Using only vacuum drying without heating and a solvent exchange still had a significant impact on the mineral composition.

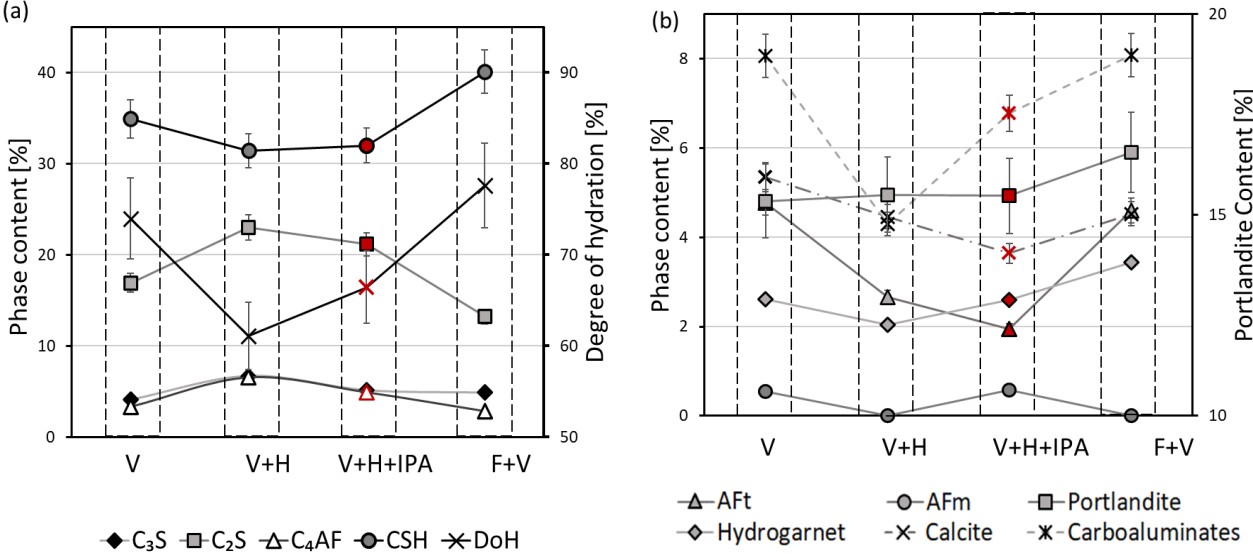

**Figure 8.** Influence of the method of the hydration stoppage on (**a**) content of cement minerals, C-S-H, and degree of hydration (calculated by TG); (**b**) content of products of hydration. The reference 2.36_IPA_1/2_V+H is marked in red.

As can be seen in Figure 8a, omitting the step of the solvent exchange from the hydration stoppage procedure had only a slight effect on the content of the anhydrous clinker minerals and C-S-H, while the DoH measured by XRD was quite lower. In the vacuum oven procedure, the content of portlandite was similar to in the solvent exchange procedure, while the content of the AFt phase and calcite was higher; the hydrogarnet and carboaluminate contents were lower, while the AFm phase disappeared completely in the vacuum oven case. According to the thermal analysis, these were also the two closest procedures with the highest difference in the range of temperatures below 105 °C, which may indicate residual free water. The presence of residual free water in the V+H sample would also explain the increase in porosity and the pore-structure coarsening observed in the MIP test.

Using a vacuum only apparently slowed down the hydration stoppage, resulting in a higher degree of hydration, a higher content of C-S-H, and a lower content of anhydrous clinker minerals. In comparison to the reference solvent exchange, vacuum drying also increased carbonation and reduced portlandite content accordingly, though the AFt and carboaluminate contents were higher. These results were quite consistent with the thermal analysis data, but the high DTG values below 105 °C and the MIP results that provide evidence for pore-structure collapse indicated that the amount of residual free water was even higher than in the case of a vacuum oven.

As Figure 8 demonstrates, when freezing in liquid nitrogen with subsequent vacuum drying was applied, the hydration stoppage was slow and resulted in a higher DoH, a higher content of C-S-H, and a lower content of anhydrous clinker minerals. As can be seen in Figure 8b, the content of portlandite, hydrogarnet, AFt, and carboaluminates was notably higher in the F+V procedure than in a solvent exchange one. The carbonation was also higher in the F+V sample, but the AFm phase was absent. These results, except carbonation, were consistent with those of thermal analysis. As in the case of vacuum drying, the high DTG values below 105 °C and the MIP results indicated that the amount of residual free water was quite significant.

### 3.4. Isothermal Calorimetry

The efficiency of each solvent might also be considered as a function of the time required for the water replacement by the solvent, i.e., the time needed to stop the hydration and also the heat release due to hydration. A heat flow chart of the samples, having a similar sample piece size and composition and apparently a similar pore structure, after 7 days of hydration and the addition of different solvents, is shown in Figure 9. It can be seen in Figure 9 that all the solvents eventually stopped hydration, which was revealed by the zero heat flow and the constant heat of hydration. In determining the time to stop the hydration, 99.5% of the final constant heat value was considered. Alternatively, less than 0.5% of the initial heat flow can be used as the hydration stoppage criterion, though both criteria resulted in similar hydration stoppage time values in our case. It should be noted that the immediate jump in heat flow at the time of replacement of calorimeter ampule is associated with both thermal equilibration and the heat released by the interaction of the solvent and hydrating cement paste.

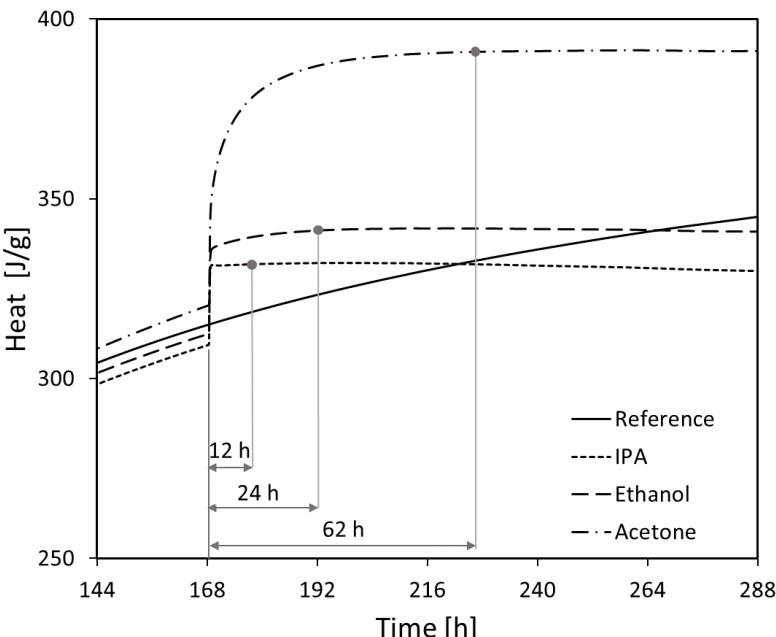

**Figure 9.** Heat flow of cement paste immersed with different solvents after 7 days (168 h) of hydration. All samples have a similar sample piece size.

The shortest time, i.e., the highest efficiency, was demonstrated by isopropyl alcohol, which stopped the reaction within approximately 40 min. The procedure of solvent replacement by ethanol required slightly above 6 h to stop the hydration. For acetone, it required almost 35 h to stop the hydration. Overall, based on the obtained results, isopropyl alcohol can be considered as the most suitable solvent for the fast and delicate stopping of hydration. Thus, isothermal calorimetry is well suited for the determination of hydration stoppage efficiency, though this was not previously reported in the literature.

### 3.5. Comparison of Results

The comparison of the results obtained by the different experimental methods, with regard to the methods of hydration stoppage, is of special interest. The comparison of the results obtained by XRD and TGA is given in Figures 10 and 11. The degree of hydration by XRD and TGA is depicted in Figure 10a. The degree of hydration calculated from isothermal calorimetry ± 5% is given in Figure 10a, with the green circle for reference. It should be emphasized that the approach to determine the degree of hydration by XRD and TGA was completely different. The approach of the XRD method to determine the degree of hydration was based on the residual cement minerals, while the TGA approach

was based on the bound water being measured by mass loss. In this regard, TGA had two parameters that affected the result. The first as the calculation of bound water, which was discussed in detail in the Section 2.3.2. The second was the temperature at which the mass loss was taken. Generally, two approaches exist to set this temperature: before (550 °C) or after (950 °C) the decarbonation step. Thus, for TGA, the results based on ignition temperatures of 550 °C (blue color) and 950 °C (magenta color) are given for comparison in Figure 10. It can be seen that the DoH calculated using the TGA obtained at 950 °C has a better agreement with the XRD results shown in Figure 10a. This indicates that some of the bound water was probably still released after 550 °C.

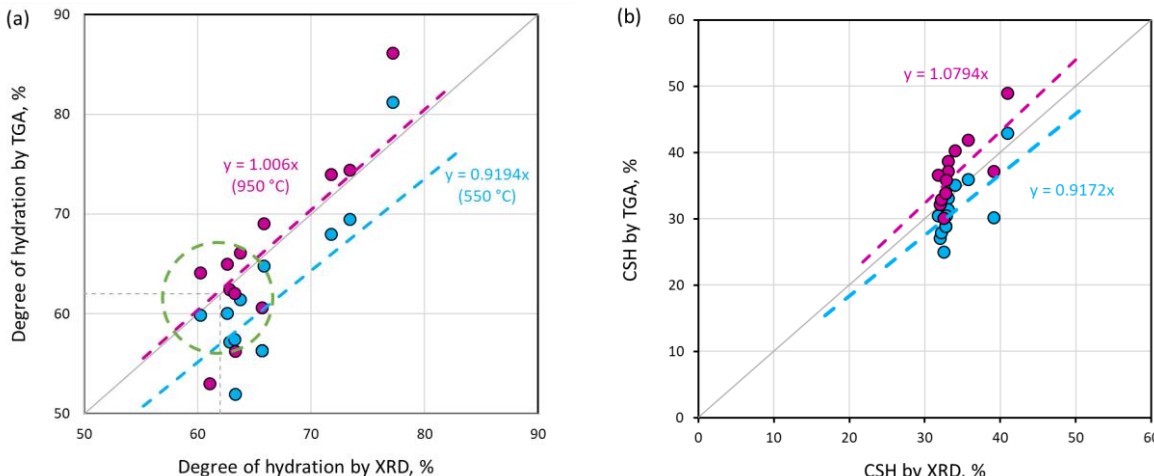

**Figure 10.** Comparison of the results obtained by XRD and TGA: (**a**) degree of hydration (green circle designates the results of isothermal calorimetry ± 5%); (**b**) C-S-H content.

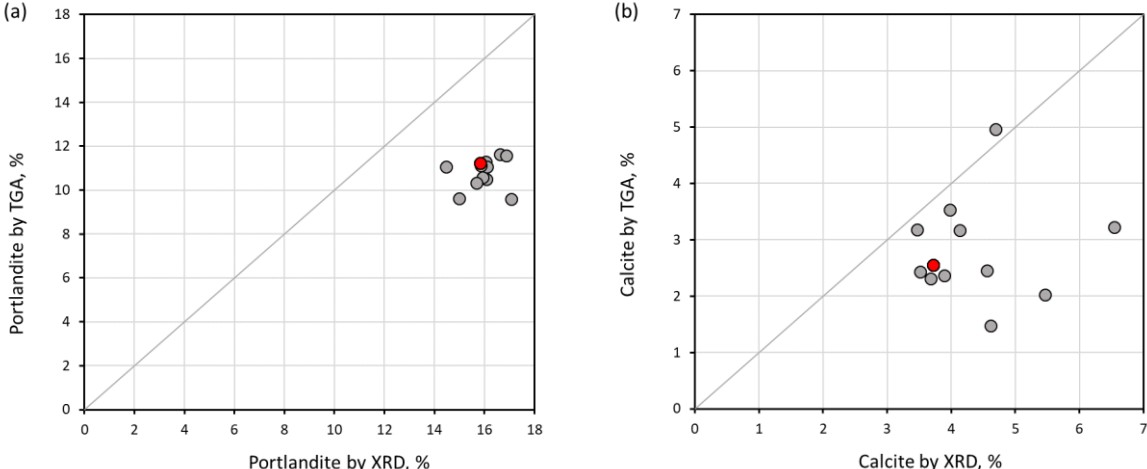

**Figure 11.** Comparison of the results obtained by XRD and TGA: (**a**) portlandite content; (**b**) calcite content.

It can be seen in Figure 10b that the calculated C-S-H content exhibited a good correlation between XRD and TGA, regardless of the ignition temperature. Again, the calculation of C-S-H by XRD and TGA uses completely different approaches. The XRD method uses an external standard for the determination of the amorphous C-S-H phase. The TGA method is based on the bound water being calculated based on a mass loss either at 550 or 950 °C. The mass losses associated with crystalline phases were evaluated and reduced from the total mass loss in order to calculate the bound water associated with the C-S-H phase only. As can be seen in Figure 10b, the C-S-H calculated based on mass loss at 950 °C gave values slightly higher than those by XRD, while the C-S-H calculated based on mass loss at 550 °C gave values slightly lower than those by XRD. This may indicate that C-S-H decomposition

continued after 550 °C but ended before 950 °C. Additionally, it can be seen that the C-S-H content by TGA was significantly affected by the method of hydration stoppage.

The content of portlandite obtained by XRD was higher by approximately 30% than that obtained by TGA, as shown in Figure 11a. Portlandite is prone to a preferred orientation due to its layered platy crystallites [40,41]. Thus, the increased portlandite content by XRD can be explained by its preferred orientation, which resulted in increased intensity of certain XRD peaks. It is worth noting that the applied method of hydration stopping had no significant influence on the amount of portlandite. In contrast to portlandite, the amount of calcite shown in Figure 11b was significantly affected by the method used for the hydration stoppage. Moreover, as in the case of portlandite, the calcite amount gained by XRD was higher than that of TGA, which may also be caused by the preferred orientation. To avoid the preferred orientation in the future, it is recommended to grind the sample into finer particles using an XRD mill.

In order to summarize all the results obtained in this study, a comparison of the methods and parameters of stopping hydration is presented in Table 4. Let us first exclude any methods showing a considerably harmful behavior to certain analytical techniques. Among them is freezing with subsequent drying under a vacuum and directly placing samples under a vacuum. Both these methods exhibited a significant change in mineral composition and pore structure. Overall, solvent replacement with subsequent gentle heating under a vacuum was proven as the most delicate method.

There was a significant influence by the sample piece size, i.e., the samples of 1.2 mm and 24 mm showed a notable change in mineral composition and degree of hydration. Therefore, the optimal sample piece size lies between 2.36 mm and 4.75 mm. The duration of solvent replacement also had a significant impact on the hydration stoppage. In the case of a very short immersion, not all the water from a sample was removed. When the samples were immersed for too long, i.e., 24 h, adverse carbonation took place. Hence, the recommended duration for the solvent exchange is between 30 and 60 min. According to the calorimetry results (see Figure 9), acetone stopped hydration slower than IPA and ethanol. Ethanol was also observed to increase the TGA peak of Aft, as can be seen in Figure 3c, probably due to the adsorption of ethanol by the AFt phase [2,12]. Therefore, only IPA can be recommended for the efficient stopping of hydration.

**Table 4.** Factors influencing hydration stoppage.

| Factor | Criteria | | | | |
|---|---|---|---|---|---|
| | Thermo-Gravimetric Analysis | X-ray Diffraction | Mercury Intrusion Porosimetry | Degree of Hydration Calculated by XRD | Agreement between Calorimetry, XRD, and TGA |
| Sample piece size: 1.2 mm; 2.36 mm; 4.75 mm; 25 mm | Samples of 1.2 mm and 25 mm exhibited increased calcite content. | Sample of 25 mm exhibits a notable increase in the content of C-S-H and Ca(OH)$_2$ and a strong reduction in AFt phase | Sample of 25 mm exhibited a notable reduction in total pore volume and augmentation of the pore entry diameter | Sample of 25 mm exhibited a notable increase in DoH | Samples of 2.36 mm and 4.75 mm |
| Duration of a solvent replacement in IPA: 15 min, 30 min, 60 min, 24 h | Immersion longer than 30 min notably increased AFt content | Immersion for 15 min reduced the amount of AFt and hydrogarnet; immersion for 24 h reduced Ca(OH)$_2$ | Immersion for 15 min drastically increased pore volume and pore entry diameter | No significant influence | Samples immersed for 30 min, 60 min, and 24 h |
| Solvent type: IPA, acetone, ethanol | Ethanol benefited augmentation of AFt peak | Ethanol and IPA increased the amount of carboaluminates | Acetone exhibited the biggest total pore volume and the pore entry diameter | No significant influence | Sample immersed in ethanol and IPA |
| Influence of drying and freezing: <br> 1. drying under a vacuum; <br> 2. freezing, and drying under a vacuum; <br> 3. heating under a vacuum; <br> 4. immersion in IPA and heating under a vacuum | Drying under a vacuum and freezing with drying under a vacuum notably increased AFt peak | Freezing and drying under a vacuum significantly increased the amount of C-S-H, calcite, Ca(OH)$_2$, and AFt; drying under a vacuum resulted in a higher amount of calcite and AFt | Freezing and drying under a vacuum and drying under a vacuum significantly reduced total pore volume, while heating under a vacuum showed the biggest pore entry diameter | Freezing and drying under a vacuum significantly increased DoH | Heating under a vacuum and immersion in IPA and heating under a vacuum |

## 4. Conclusions

The current research presents a systematic investigation of the methods and factors influencing the stoppage of the hydration of hardened cement paste. The impact of the sample piece size, soaking time in a solvent, and type of solvent is presented. As a result, a quantitative comparison between the well-known methods of hydration stoppage and their influence on the different analytic techniques, such as TGA, MIP, and XRD, is given. The main conclusions linked to each factor studied in the research can be written as follows:

- The optimal sample piece size required for hydration stoppage is ca. 2–4 mm. Pieces of smaller and larger sizes exhibit a higher content of calcium carbonate. Additionally, larger sizes of samples prevent the complete removal of free water.
- The optimal soaking time of a solvent replacement is considered to be between 30 min and 1 h. A faster solvent replacement, such as 15 min, may leave free water in the sample, which will cause capillary stresses during drying, dramatically increasing the total pore volume and decreasing the amount of ettringite. However, soaking times longer than a few hours evidently promote the carbonation of the samples.
- Ettringite is considered the most sensitive phase for the different methods of hydration stoppage.
- Direct freezing with a vacuum treatment and direct placement under a vacuum affect the pore structure of the sample the least, whereas the mineral composition is heavily affected. Solvent replacement combined with a vacuum and heating at 40 °C and direct placement under a vacuum and heating at 40 °C provide comparable and reproducible results.
- The degree of hydration and the amount of C-S-H and portlandite calculated using TGA at 950 °C have a better agreement with the XRD result, due to the remaining bound water, which is still preserved at 550 °C
- The efficiency of the different solvents employed for hydration stoppage was assessed using isothermal calorimetry. In terms of the time needed for complete hydration

stoppage, the solvents are ordered from most to least effective as follows: isopropyl alcohol > ethanol > acetone.

**Author Contributions:** Methodology, A.M. and S.Z.; Validation, A.M. and S.Z.; Formal analysis, A.M., A.G. and S.Z.; Investigation, A.M., D.K., A.G. and S.Z.; Resources, S.Z.; Data curation, A.M. and S.Z.; Writing—original draft, A.M., D.K., A.G. and S.Z.; Writing—review & editing, S.Z.; Visualization, A.M. and S.Z.; Supervision, S.Z.; Project administration, S.Z.; Funding acquisition, S.Z. All authors have read and agreed to the published version of the manuscript.

**Funding:** This research received no external funding.

**Acknowledgments:** A.M. acknowledges the Centre for Absorption in Science, Ministry of Immigrant Absorption, State of Israel, for their financial support.

**Conflicts of Interest:** The authors declare no conflict of interest.

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
