# Peer review of "A Comparative Study of Factors Influencing Hydration Stoppage of Hardened Cement Paste"

_sustainability, doi:10.3390/su15021080_

Round 1

Reviewer 1 Report

Overall a very good approach on the investigation of the methods and parameters affecting hydration stoppage of hardened cement pastes. The methodology used is adequately described (although certain issued have been identified that should be looked at). The results are clearly presented and discussed (especially section 3.5). Upon successful consideration of the issues identified on the annotated version of the document, the current research study deserves to be published in the Journal.    

Reviewer 2 Report

This paper is excellent.

Reviewer 3 Report

This paper presents a quantitative comparison of the most common hydration stopping techniques and variables like sample piece size, solvent soaking time, and solvent type have been looked at on cement paste that had been hydrated for seven days. The results show that samples larger than 4.75 mm and smaller than 2.36 mm exhibit carbonation, as do those immersed in a solvent for more than an hour. The rapid solvent replenishment causes the ettringite to decrease and the total pore volume to increase. Also, the results discuss that, the solvent substitution with subsequent moderate heating under vacuum was the most effective, and that isopropyl alcohol stopped hydration more quickly than ethanol and acetone. The overall preparation of the paper and analysis presentation is significant; however, the following comments can be considered for improving the flow and discussion:

1.      The authors mentioned lots of chemical components during the analysis; these components are not clear to all readers, therefore including an abbreviation table describing the chemical terms would be helpful.

2.      The numerical results need to be avoided in the abstract, only intro for the results and an overall idea of the findings

3.      Clear explanation of some figures needs to be included as well as the figures presentation of data; the relation between some figures and their captions are not clear, for example figure 3. Also, the authors need to avoid long descriptive captions, they can be confusing to readers/

4.      The introduction and literature review about previous research is lacking how this research is building on them through its findings.

5.      A discussion needs to be provided relating this work with previous research and recommending road for the future works

6.      The paper is lacking discussion about the different factors included in the analysis, such as the sample piece size, soaking time in a solvent, and its type; why these were chosen with their impact on the study, what about other factors.

7.      The conclusion needs to be written in a paper format; the large number of bullet points included right now is more recommended for report formatting.

8.      The reference section needs to be revised to have one format for all the references.

Round 2

Reviewer 3 Report

Reconsider the comments given in te 1st round

Round 3

Reviewer 3 Report

They addressed most of the comments it can be accepted now